# Analysis and Forecast of Land Use and Carbon Sink Changes in Jilin Province, China

**Mengqi Wei [1], Chong Du [1,*] and Xuege Wang [2,*]**

[1] College of Water Resources and Electric Power, Heilongjiang University, 74 Xuefu Road, Harbin 150080, China; w2531661590@163.com

[2] College of Water Resources and Environmental Engineering, Changchun Engineering College, Changchun 130012, China

* Correspondence: duchong@hlju.edu.cn (C.D.); 13504314997@163.com (X.W.); Tel.: +8613633661083 (C.D.)

**Abstract:** Based on the land use data changes in Jilin Province in 2011, 2014, 2017, and 2020, this paper analyzes the land use changes during 2011–2020 through the land use transfer matrix, calculates the changes in carbon sinks of recent years, and then uses the CA–Markov model to predict the land use types and carbon sinks in Jilin Province in 2030 and discusses the driving factors. The results show that cultivated land and forest land are the two major land use types in Jilin Province, and the area of cultivated land, water bodies, and artificial ground in the province increased from 2011 to 2020; the increased area of artificial ground was mainly converted from cultivated land, accounting for 70.34% of the total converted area. The area of forest land is mainly converted along with the area of cultivated land, and grassland is mainly converted to arable areas, accounting for 84.96% of the total converted area. Water bodies and wasteland are mainly converted to cropland and artificial ground, and the area of artificial ground undergoing transfer is smaller. The change in carbon sinks mainly comes from woodland carbon sinks and grassland carbon sinks. In 2030, compared with 2020, the area of woodland, grassland, and wasteland and the corresponding carbon sink is predicted to decrease, among which the area and carbon sink of woodland decrease the most. The factors for land use type change include the slope factor, road factor, township center, and socio-economic drivers.

**Keywords:** land use; carbon sink; CA–Markov model





## 1. Introduction

All forms of life on Earth are derived from the land, and as material foundations for human survival and development, development and human activities are inseparable from the land and will ultimately be reflected in land use. Land use may affect the carbon sink of an area, and thus influence the carbon balance. The demand for social development causes people to transform and use the land according to its various properties, and the land use pattern is changed [1], and the carbon sink is also altered. Human beings fundamentally occupy the dominant position in terms of land use, and the rapid development of society, human's own needs, and changes in the ecological environment are all inextricably bound up with changes in land use. The local natural environment [2–4] is negatively affected by land use change. The expansion of urban construction comes at the expense of natural vegetation cover and is therefore considered by many countries to be an irreversible change in land use type [5]. Cities occupy cropland and cropland occupies forest land area. Additionally, since land use is the activity that connects people most directly with the natural world, regional land use change has significant impacts on human activity, climate, and the environment as a whole [6]. Consequently, one of the current key areas of research on global change is land use change research. We should treat the relationship between humans and nature properly and combine nature organically, together with science and technology as well as human power when considering problems, to properly understand the relationship between man and Earth. The essence of addressing climate

change is, on the one hand, consistent with China's developmental trajectory, and low carbon development will foster fundamental improvements in the quality of China's ecological environment; on the other hand, the achievement of carbon peak and carbon neutrality is also the core of today's global competitiveness. Technological and economic competition underlie carbon neutrality. This will lead to the research into and development of a new generation of technology in a variety of countries, and the world will be entering an era of technological change in energy, industry, transportation, and construction in the next few years. Carbon neutrality is going to be a key core technology and a time of strategic development opportunity for the world, and we need to take that opportunity.

Over the past few decades, many researchers have investigated changes in the spatial patterns of land use in different regions [7–9], scales [10–12], and scenarios [13–15]. In line with the UN 2030 Agenda for Sustainable Development, Estoque et al. [16] carried out an analysis of the relationship between land use, population, and social growth. Kuwari et al. [17] used remote sensing technology to systematically assess urban land use changes due to the construction of the Ras Laffan oil field port in northern Qatar. Cui et al. [18] used the emission factor approach proposed by the IPCC to study land use/vegetation. An examination of changes between 1990 and 2015 in the Beijing–Tianjin–Hebei urban agglomeration was carried out. The impact of land use change on environmental change [19–22] has gradually received academic attention. Rational land use is an efficient way to grow carbon sinks; previous land use planning prioritized economic gains, resulting in increased pressure on ecosystem conservation and inadequate land use [23]. Population growth and economic development have led to a dramatic change in land use [24], especially with the continued expansion of cities leading to an increase in building lots.

Jilin Province is represented as the northeast region of China. With the rapid social and economic development, the region's intensity of land usage and development began to rise. Scholars have researched land use changes in the region, although the majority of their research has concentrated on changes in drivers and ecological repercussions [25–29], and fewer involve studies on land use and carbon sink change prediction. Therefore, based on the study of land use in Jilin Province, analyzing and predicting the developmental changes in land use change and carbon sinks can provide decision making suggestions for other cities. In 2014, China released the first domestic National Plan for Responding to Climate Change (2014–2020), which once again clarified the action objectives for responding to climate change. This study takes Jilin Province as an object and examines the process of and trend in land use change by examining the differences in the spatial and temporal distribution of different land uses in 2011–2014 (before releasing the plan) and 2017–2020 (after releasing the plan). In 2011–2014 and 2017–2020 the differences in the spatial and temporal distribution of different land uses reflect the process of and trend in land use changes in Jilin Province. The land use types and carbon sinks were calculated for different periods, and finally, the prediction of land use types and carbon sinks in 2030 was carried out by the CA–Markov model, which initially explored the driving factors of land use changes and their socio-economic consequences. This is crucial for understanding the evolutionary patterns of land use in different cities, promoting the rational use of land resources, and facilitating sustainable economic growth.

## 2. Study Area and Methods

### 2.1. Overview of the Study Region

Jilin Province, which covers an area of 187,400 km$^2$, is located in Northeast China's hinterland (Figure 1), with a length of 769.62 km from east to west and a width of 606.57 km from north to south, south of Liaoning Province, west of Inner Mongolia Autonomous Region, north of Heilongjiang Province, and east of the Russian Federation, with the Democratic People's Republic of Korea to the southeast. Jilin Province has obvious differences in landform, with the terrain sloping from southeast to northwest, showing obvious characteristics of highland in the southeast and lowland in the northwest. Taking the central Dahei Mountain as the boundary, two main geomorphic regions can be distinguished: the

eastern mountains and the central and western plains. The eastern mountainous area is divided into the low mountainous area of Changbai Mountain and the low mountainous hilly area, while the central and western plains are divided into the central tableland plain area and the western meadows, lakes, wetland, and barren areas; the central plains consist of a tableland plain area. Of the total area, mountains account for 36%, plains account for 30%, tablelands and others account for 28.2%, and the rest are hills.

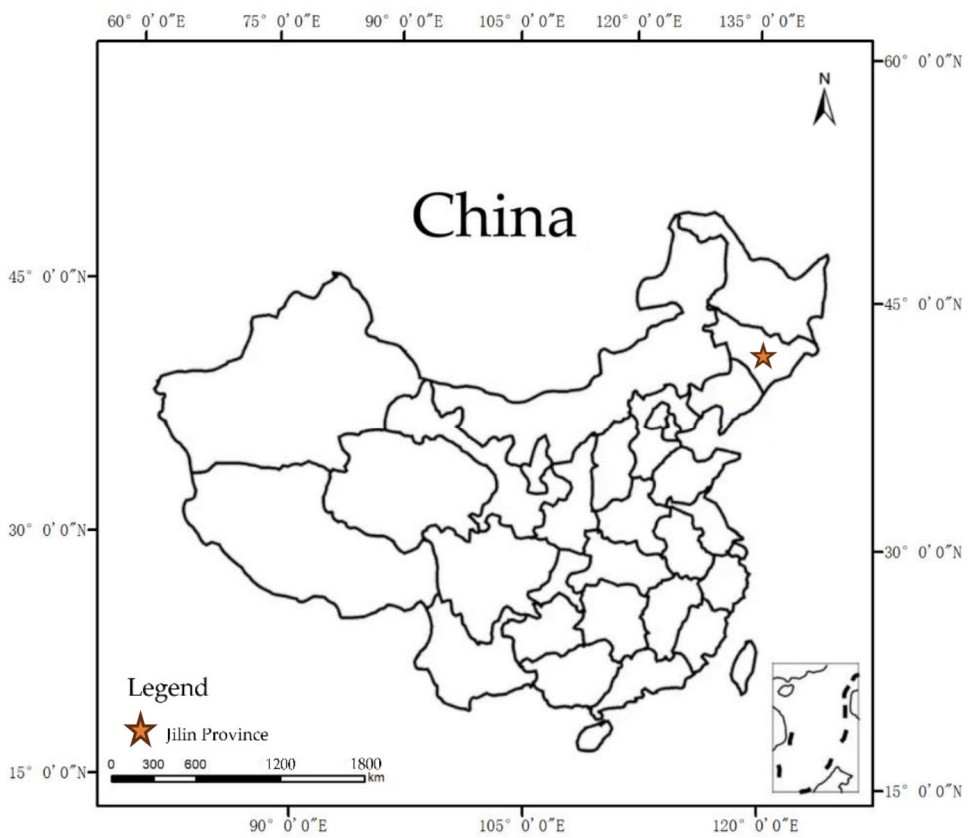

**Figure 1.** Location Map of Jilin Province.

### 2.2. Data Sources and Pre-Processing

The designated years of land use data are 2011, 2014, 2017 and 2020. The data were obtained from the Resource and Environmental Science and Data Center (REDC) of the Chinese Academy of Sciences (CAS), Jilin-1 satellite, and Landsat 8 with a resolution of 30 m × 30 m. The land use types of the study site were classified and extracted by using ARCGIS and ENVI, using different satellites and time periods, into six categories of cropland, woodland, grassland, water, wasteland, and artificial land.

### 2.3. CA–Markov Model

The CA–Markov model is an integration of Cellular Automata (CA) and the Markov model to simulate the dynamics of land use structure change. The CA model is based on the premise that land use change occurs through the interaction of different land use types in a region, with the land use types represented by the central cell change under the effect of the land use types of the domain cell, while the Markov chain controls the time change through the conversion matrix of land use types. These are represented by the center tuple changes under the action of the domain tuple land use types; Markov chains are used to control the temporal changes through the conversion matrix of land use types.

(1)    CA model

Cellular Automata, first proposed by von Neumann and dating back to the 1950s, is essentially a scientific method for studying spatiotemporal evolution based on discontin-

uous spatiotemporal dynamics models, which are discrete in both temporal and spatial states. A meta cellular automaton is composed of the following four parts:

① Cell. Each meta cell is a unit that has a corresponding information state at each fixed moment. For example, the state of a beta cell is a certain land use category.

② Metacell space (lattice). This is a collective of combined tuples. Generally, two-dimensional tuple automata are arranged in triangular, quadrilateral, or hexagonal structures.

③ Neighborhoods. These are simply the neighbors of a tuple, an ensemble of neighboring tuples surrounding the tuple. Common types are Von Neumann, Moore, and extended Moore: Von Neumann means that the domain of a tuple consists of four neighboring tuples, one above, one below, one to the left, and one to the right; Moore means that the domain of a tuple consists of eight tuples around its perimeter. Moore type means that the domain of a tuple has a radius greater than or equal to 2 for all surrounding tuples.

④ Rules. This refers to a rule-determined state transfer model, which can derive the state of a tuple at the next moment based on the state of the tuple at this moment and its domain.

(2)    Markov model

This is a method of predicting the probability of the occurrence of events created by the former Soviet mathematician Markov in the 1940s, so it is called the Markov Chain; it can predict the trend in changes in future moments, and is a kind of spatial probabilistic model based on the grid. Its main parts are as follows:

① Markov process. This means that in the process of state transfer, the current state is related to the state of the previous moment, and has nothing to do with the situation before the current state, i.e., there is no posteriority.

② State transfer probability. That is, from a certain state to the next moment of the state transfer probability.

③ State transfer probability matrix. That is all the various forms that may occur in a change, i.e., all transfer probabilities.

④ Calculate the state transfer probability matrix.

(3)    Combination of CA and Markov.

The CA model can be good and effective in mimicking the spatial changes of the system, in that it has excellent spatial arithmetic ability and a strong spatial concept, but it cannot analyze the overall cellular state; the Markov model lacks the ability in the spatial simulation of the changes, but it has a better prediction ability in terms of quantity. Each of them has its characteristics in spatial simulation, but both of them have limitations. Combining the two organically and complementing their strengths to become the CA–Markov model not only achieves more accurate prediction in quantity but can also obtain more intuitive prediction results at the spatial layout level.

The CA and Markov model expression equations are as follows:

$$S_{t+1} = \int (S_t, N) \tag{1}$$

$$P_{ij=}\begin{bmatrix} P_{11} & P_{12} & \cdots & P_{1n} \\ P_{21} & P_{22} & \cdots & P_{2n} \\ \vdots & \vdots & \cdots & \vdots \\ P_{n1} & P_{n2} & \cdots & P_{nn} \end{bmatrix} \tag{2}$$

$$P_{ij}^{(n)} = \sum_{k=1}^{N} P_{ik} P_{kj}^{(n-1)} = \sum_{k=1}^{N} P_{ik}^{(n-1)} P_{kj} \tag{3}$$

where $S$ is the space of cells; $t$ denotes the moment; $N$ is the domain of each cell; $f$ is a regular function of the evolution of cell states; and $P_{ij}$ is the probability of moving from one state to the next. All $n$ morphologies may occur during the change.

### 2.4. Land Use Transfer Matrix

The matrix of land use migration can reflect the degree of interconversion between various categories in a region at a certain time, which can visually reflect the dynamic alterations in land use patterns and the spatial and temporal evolution process of land use patterns.

$$S_{ij} = \begin{bmatrix} S_{11} & S_{12} & \cdots & S_{1n} \\ S_{21} & S_{22} & \cdots & S_{2n} \\ \vdots & \vdots & \cdots & \vdots \\ S_{n1} & S_{n2} & \cdots & S_{nn} \end{bmatrix} \tag{4}$$

where $n$ is the number of land use types, in this study $n = 6$; $S_{ij}$ denotes the area transferred from $i$ to $j$. $S_{11}$, $S_{22}\ldots_{nn}$ denotes the area where there is no change in land use type.

### 2.5. Land Use Change Rate

The rate of change in land use area is an indicator of land use change in the study area over time. It can reflect the trajectory of the rate of change in the study area and can be subdivided into relative rate of change and net rate of change [30]. The relative rate of change represents the change in land use area relative to the initial stage, while the net rate of change represents the average annual rate of change. Analysis of the relative rate of change and the net rate of change of land use area in the study area resulted in a multidimensional systematic analysis.

$$N_c = \frac{U_b - U_a}{U_a} \tag{5}$$

$$R_s = \left[ \sqrt[T]{\frac{U_b}{U_a}} - 1 \right] \times 100\% \tag{6}$$

where $N_c$ is the relative change rate of the area, $R_s$ is the net change rate, $U_a$ and $U_b$ are the areas at the beginning and conclusion of the study period, and T is the duration of the research. Regardless of the mutual transfer in and out for all types, the altered regions in question are all net change regions.

### 2.6. Carbon Sink Value

Through the land use classification system and the area's genuine land use, Jilin Province's land use categories were divided into six first-level classifications: cultivated land, grassland, forest land, water body, artificial ground, and wasteland. Among them, cropland, grassland, and woodland are the main sources of terrestrial carbon sink, and the carbon sink of each land use type is calculated by the formula:

$$C_a = \sum T_i = \sum S_i \times H_i \tag{7}$$

where $C_a$ denotes carbon sequestration; $T_i$ denotes the carbon sequestration capacity of the ith land type in the study area (Table 1); and $S_i$ denotes the area of the ith category of terrain. The ith land type has a carbon sequestration coefficient of $H_i$ [31–34].

$$C = C_a \times \frac{44}{12} \tag{8}$$

$C$ in the formula is the carbon sink, and the final carbon sink in this paper is the carbon dioxide uptake, so the carbon uptake should be multiplied by the conversion factor of carbon and carbon dioxide 44/12.

**Table 1.** Land use type: carbon sequestration capacity (tC/ ha·a).

| Land Use Types | Carbon Absorption Capacity | Land Use Types | Carbon Absorption Capacity |
|---|---|---|---|
| forest land | 0.542–0.585 | water body | 0.069–0.074 |
| grass land | 0.024–0.027 | waste land | 0.0039–0.0042 |
| cultivated land | 0.0048–0.0053 | artificial ground | 0.0049–0.0051 |

## 3. Results

### 3.1. Land Use Changes

Figure 2 depicts the spatial distribution of land use in Jilin Province over time. Table 2 and Figure 3 illustrate the areas and proportions of various land use categories during the study period. The maximum proportion of cultivated land indicates that it is the dominant form of land in the region, followed by forest land area. The order of land area was: cultivated land > forest land > artificial ground > grass land > water body > waste land. As shown in Figure 3, the proportion of cultivated land reaches 46.29%, 46.89%, 47.12%, and 46.94% from 2011 to 2020, respectively, showing an increase and then a decrease with time. In contrast, forest land shows a trend of decreasing and then increasing, with percentages of 43.29%, 42.93%, 42.95%, and 43.15%, respectively. The percentage of wasteland and grass decreased from 3.73% and 1.30% to 2.69% and 1.01%, respectively, while the percentage of water body and artificial ground increased from 1.37% and 4.02% to 1.46% and 4.75%, respectively.

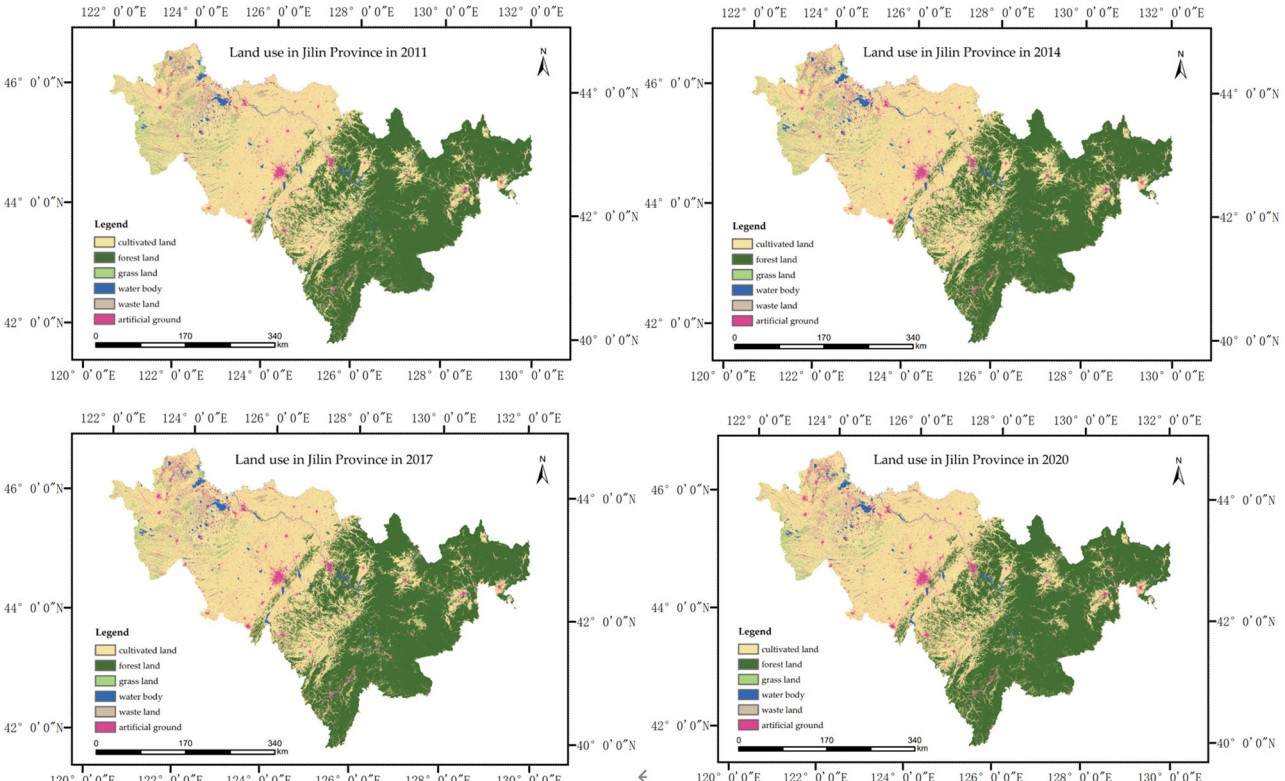

**Figure 2.** Land use spatial pattern in Jilin Province from 2011 to 2020.

**Table 2.** Area of land use type in Jilin Province from 2011 to 2020 (km$^2$).

| Year | Cultivated Land | Forest Land | Grass Land | Water Body | Waste Land | Artificial Ground |
|------|-----------------|-------------|------------|------------|------------|-------------------|
| 2011 | 88,316.84 | 82,596.73 | 7120.98 | 2611.40 | 2472.70 | 7664.18 |
| 2014 | 89,456.54 | 81,898.20 | 6022.22 | 2890.11 | 2233.91 | 8283.15 |
| 2017 | 89,897.13 | 81,951.06 | 5408.37 | 2772.95 | 2051.16 | 8704.77 |
| 2020 | 89,555.70 | 82,320.81 | 5136.98 | 2782.87 | 1917.81 | 9071.03 |

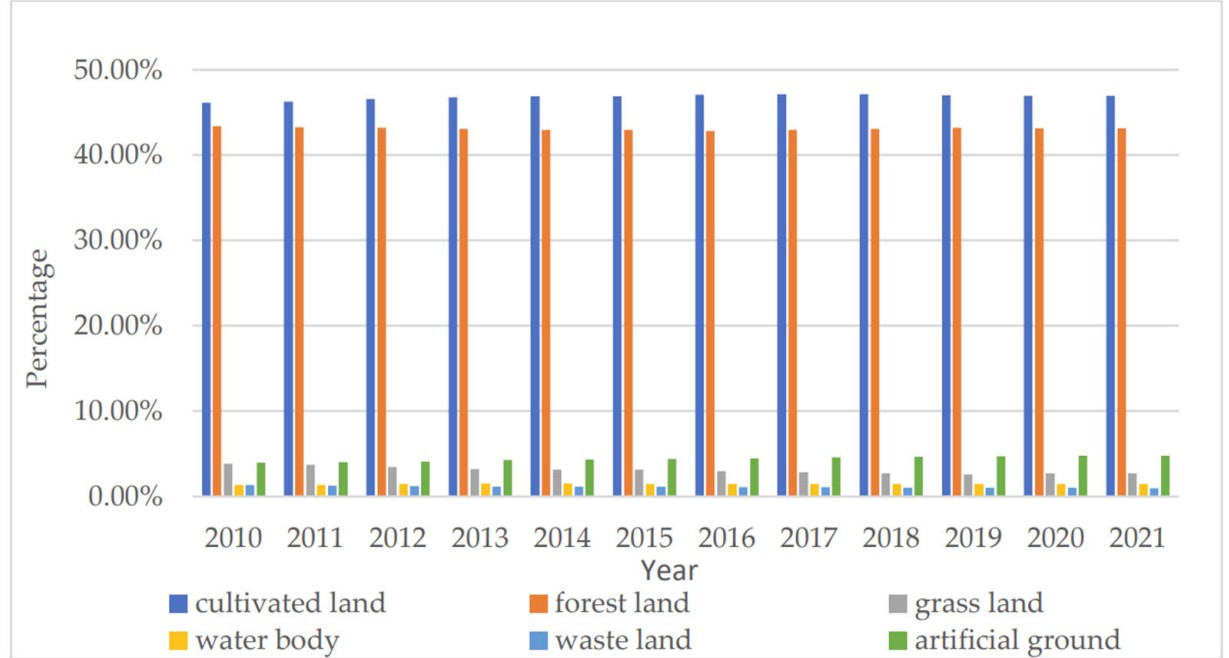

**Figure 3.** Changing proportions of land use over time.

Table 2 also shows that, since 2011, the area of grassland and wasteland has shown a decreasing trend, with the area of grassland decreasing to a more significant extent; the construction land area has exhibited a significant upward trend, and the area of water body, cropland, and woodland has been fluctuating. Table 3 shows the rate of change of all types in different periods. From 2011 to 2014, the area of woodland, grassland, and wasteland decreased, among which the area of grassland decreased the most significantly, with 1098.76 km$^2$. This was followed by woodland and wasteland, which decreased the least, with only 238.79 km$^2$. By eradicating the impact of each land use type's base area, the relative rate of change and the net rate of change of the grassland area were the largest, with −15.43% and −5.43%, respectively. The area of wasteland decreased by 238.79 km$^2$, with relative and net change rates of −9.66% and −3.33%, respectively. The relative rate of change and net rate of change of the water body area were 10.67% and 3.44%, respectively, and the relative rate of change and net rate of change of artificial ground area were close to those of the water body area. From 2014 to 2017, except for water body and forest land, the area of all land types had the same trend in change as the previous period. The forest land area changed from decrease to increase, while the area of the water body changed from increase to decrease. Compared to 2011−2014, the net rate of change for every land use type fell in 2017−2020, with the most significant change in the net rate of change in the area of grassland. Similarly, the relative rate of change for 2017−2020 was lower than that for 2011−2014, with the grassland area decreasing the most and the artificial ground area increasing the most.

**Table 3.** Changes in various land use types in Jilin Province from 2011 to 2020.

| Time Interval | Index | Cultivated Land | Forest Land | Grass Land | Water Body | Waste Land | Artificial Ground |
|---|---|---|---|---|---|---|---|
| 2011–2014 | Area variation (km$^2$) | 1139.70 | −698.53 | −1098.76 | 278.71 | −238.79 | 618.97 |
| | Relative change rate (%) | 1.29 | −0.85 | −15.43 | 10.67 | −9.66 | 8.08 |
| | Net change rate (%) | 0.43 | −0.28 | −5.43 | 3.44 | −3.33 | 2.62 |
| 2014–2017 | Area variation (km$^2$) | 440.59 | 52.86 | −613.85 | −117.16 | −182.75 | 421.62 |
| | Relative change rate (%) | 0.49 | 0.06 | −10.19 | −4.05 | −8.18 | 5.09 |
| | Net change rate (%) | 0.16 | 0.02 | −3.52 | −1.37 | −2.80 | 1.67 |
| 2017–2020 | Area variation (km$^2$) | −341.43 | 369.75 | −271.39 | 9.92 | −133.35 | 366.26 |
| | Relative change rate (%) | −0.38 | 0.45 | −5.02 | 0.36 | −6.50 | 4.21 |
| | Net change rate (%) | −0.13 | 0.15 | −1.70 | 0.12 | −2.22 | 1.38 |
| 2011–2020 | Area variation (km$^2$) | 1238.86 | −275.92 | −1984.00 | 171.47 | −554.89 | 1406.85 |
| | Relative change rate (%) | 1.40 | −0.33 | −27.86 | 6.57 | −22.44 | 18.36 |
| | Net change rate (%) | 0.15 | −0.04 | −3.56 | 0.71 | −2.78 | 1.89 |

Throughout the study period, the area of grassland, water body, and wasteland declined and continued to decrease, the area of built-up land increased, and the area of cropland and forest land was in a fluctuating state. Overall, the net rates of change from high to low were: construction land, water body, cropland, forest land, wasteland, and grassland.

*3.2. Land Use Transfer Changes*

The matrix of land use migration for 2011–2014, 2014–2017, 2017–2020, and 2011–2020 was used to evaluate the transition between various land use categories in Jilin Province from 2011 to 2020 (Tables 4–7). During the period 2011–2014 (Table 4), the largest land use area was transferred out of cultivated land, of which 36.14%, 29.59%, and 23.38% of the total transferred area was transferred to forest land, grassland, and artificial land surface, respectively. Most of the forest land and grassland was transferred to cropland, accounting for 98.83% and 85.5%, respectively. Water body and wastelands were also mainly converted to cropland, accounting for 57.28% and 40.46% of the total area transferred, respectively.

**Table 4.** Matrix of land use transfer in Jilin Province from 2011 to 2014.

| Land Use Type | | 2014 | | | | | |
|---|---|---|---|---|---|---|---|
| | | Cultivated Land | Forest Land | Grass Land | Water Body | WASTE LAND | Artificial Ground |
| 2011 | cultivated land | 86,295.12 | 761.39 | 623.19 | 229.15 | 0 | 492.54 |
| | forest land | 1448.37 | 81,177.83 | 0 | 0 | 0 | 17.13 |
| | grass land | 1543.42 | 4.39 | 5327.38 | 59.12 | 125.04 | 73.26 |
| | water body | 101.61 | 1.82 | 3.44 | 2455.69 | 11.42 | 59.11 |
| | waste land | 151.74 | 0.00 | 77.65 | 75.67 | 2098.70 | 70.00 |
| | artificial ground | 0 | 0 | 0 | 91.02 | 0 | 7590.53 |

**Table 5.** Matrix of land use transfer in Jilin Province from 2014 to 2017.

| Land Use Type | | 2017 | | | | | |
| --- | --- | --- | --- | --- | --- | --- | --- |
| | | Cultivated Land | Forest Land | Grass Land | Water Body | Waste Land | Artificial Ground |
| | cultivated land | 87,611.38 | 770.61 | 786.42 | 67.10 | 0 | 305.90 |
| | forest land | 711.65 | 81,222.65 | 0 | 0 | 0 | 11.13 |
| 2014 | grass land | 1320.41 | 3.30 | 4529.45 | 4.14 | 121.91 | 52.74 |
| | water body | 150.33 | 2.13 | 11.47 | 2660.90 | 29.23 | 56.67 |
| | waste land | 187.46 | 0 | 89.82 | 14.11 | 1901.60 | 42.10 |
| | artificial ground | 10.92 | 0.11 | 1.61 | 44.39 | 2.43 | 8258.18 |

**Table 6.** Matrix of land use transfer in Jilin Province from 2017 to 2020.

| Land Use Type | | 2020 | | | | | |
| --- | --- | --- | --- | --- | --- | --- | --- |
| | | Cultivated Land | Forest Land | Grass Land | Water Body | Waste Land | Artificial Ground |
| | cultivated land | 87,562.55 | 982.61 | 1068.79 | 65.00 | 0 | 303.25 |
| | forest land | 610.88 | 81,378.23 | 0 | 0 | 0 | 9.59 |
| 2017 | grass land | 1133.46 | 5.62 | 4025.71 | 6.34 | 208.63 | 36.41 |
| | water body | 59.82 | 1.25 | 14.93 | 2655.61 | 22.13 | 36.96 |
| | waste land | 272.29 | 0 | 36.89 | 18.84 | 1688.73 | 36.02 |
| | artificial ground | 0 | 0 | 0 | 54.78 | 0 | 8671.95 |

**Table 7.** Matrix of land use transfer in Jilin Province from 2011 to 2020.

| Land Use Type | | 2020 | | | | | |
| --- | --- | --- | --- | --- | --- | --- | --- |
| | | Cultivated Land | Forest Land | Grass Land | Water Body | Waste Land | Artificial Ground |
| | cultivated land | 83,701.96 | 1795.29 | 1528.72 | 248.55 | 42.68 | 1084.00 |
| | forest land | 2001.94 | 80,557.72 | 38.77 | 3.61 | 0.01 | 41.29 |
| 2011 | grass land | 3142.20 | 9.98 | 3433.23 | 43.09 | 335.42 | 167.72 |
| | water body | 177.89 | 4.42 | 10.21 | 2307.40 | 30.19 | 102.95 |
| | waste land | 602.17 | 0.02 | 133.43 | 84.44 | 1508.75 | 145.09 |
| | artificial ground | 10.92 | 0.11 | 1.61 | 113.36 | 2.43 | 7553.11 |

During the period 2014–2017 (Table 5), the surface of transfer from grassland to cropland decreased, to 1320.41 km$^2$. The form of transfer between water body and wasteland did not change much. The changes in the area between cultivated land and woodland and between water body and wasteland are mainly reflected in their mutual transformation. The artificial ground is primarily transformed into a body of water, although the transferred area is minor.

During the period 2017–2020 (Table 6), the area transferred to and from cropland remained the largest, being mainly transferred to forest land and grassland, and accounting for 40.61% and 44.17%, respectively, with grassland and cropland in a mutual transformation relationship; this wsa followed by transfer to artificial ground, accounting for 12.53% of the total transferred area, and the area transferred least to the water body. During this period, the area of forest land was mainly converted to cropland, accounting for 98.45% of the total transferred area. The area of water body converted into other types of land decreased significantly. The wasteland was mainly converted to cultivated land, with a conversion area of 272.29 km$^2$. The artificial ground was transferred only to the water body, with an area of 54.78 km$^2$.

Looking at the entire study period (Table 7), the increased area of artificial ground was mainly converted from cropland, accounting for 70.34% of the total converted area. The

forested area was mainly converted to cropland area, and grassland was mainly converted to arable area, accounting for 84.96% of the total converted area. The water body and wasteland were mainly converted to cultivated land and artificial ground, and the area of artificial ground that underwent conversion was smaller.

### 3.3. Carbon Sink Change

For different land use types, from the time series (Table 8), the carbon sinks in Jilin Province in 2011, 2014, 2017, and 2020 were 18,058,400 $tCO_2$, 17,875,200 $tCO_2$, 17,879,900 $tCO_2$ and 17,959,300 $tCO_2$, respectively. During the period 2011–2014, for the carbon sinks of cultivated land, water body, and artificial ground, there was a small increase in the carbon sink of the land surface. The carbon sinks of woodland, grassland, and wasteland decreased, among which woodland decreased the most, with a reduction of 149,800 $tCO_2$. The annual carbon sink decrease was 148,000 $tCO_2$. The carbon sinks of all types of land use types fluctuated little during 2014–2017, and the total carbon sink only increased by 0.47 million $tCO_2$. During 2017–2020, the carbon sink changes were mainly for woodland. The change in carbon sink was mainly from the woodland carbon sink and grassland carbon sink, which decreased by 59,200 $tCO_2$ and 15,200 $tCO_2$, respectively, and the total carbon sink decreased by 66,000 $tCO_2$ in the whole study period.

**Table 8.** Analysis of carbon sinks in Jilin Province from 2011 to 2020 ($10^4$ $tCO_2$).

| Year | Cultivated Land | Forest Land | Grass Land | Water Body | Waste Land | Artificial Ground | Total |
|---|---|---|---|---|---|---|---|
| 2011 | 16.19 | 1771.70 | 5.48 | 7.09 | 0.45 | 1.41 | 1802.32 |
| 2014 | 16.40 | 1756.72 | 4.64 | 7.84 | 0.41 | 1.52 | 1787.52 |
| 2017 | 16.48 | 1757.85 | 4.16 | 7.52 | 0.38 | 1.60 | 1787.99 |
| 2020 | 16.42 | 1765.78 | 3.96 | 7.55 | 0.35 | 1.66 | 1795.72 |

### 3.4. Land Use and Carbon Sink Projections

The current land use map of Jilin Province in 2020 is used as the basis, and the CA–Markov module in IDRISI Selva is applied to predict the land use structure of Jilin Province in 2030, using the land use transfer probability (Table 9) and the land use change suitability atlas from 2011 to 2020 as the conversion rules.

**Table 9.** Probability matrix of land use type transition of Jilin from 2011 to 2020.

| | Cultivated Land | Forest Land | Grass Land | Water Body | Waste Land | Artificial Ground | Transfer Probability |
|---|---|---|---|---|---|---|---|
| Cultivated land | 64.15% | 13.14% | 11.25% | 2.26% | 0.44% | 8.75% | 35.84% |
| Forest land | 24.96% | 73.94% | 0.52% | 0.05% | 0.00% | 0.53% | 26.06% |
| Grass land | 52.81% | 0.17% | 38.90% | 0.72% | 4.71% | 2.68% | 61.09% |
| Water body | 14.40% | 0.32% | 0.84% | 73.94% | 2.12% | 8.38% | 26.06% |
| Waste land | 30.32% | 0.00% | 7.62% | 4.47% | 50.24% | 7.35% | 49.76% |
| Artificial ground | 1.48% | 0.04% | 0.20% | 14.64% | 0.35% | 83.30% | 16.71% |
| Transfer probability | 188.12% | 87.61% | 59.33% | 96.08% | 57.86% | 110.99% | |

As can be seen from the results (Table 10, Figure 4), the area of woodland, grassland, and wasteland and the corresponding carbon sinks decreased, compared to 2020, with the area and carbon sinks of woodland decreasing the most, at 770.74 $km^2$ and 165,300 $tCO_2$, respectively. The cultivated land, water body, and artificial ground showed a small increase in carbon sinks. The overall carbon sink was 17,805,000 $tCO_2$, a decrease of 156,700 $tCO_2$ from 2020.

**Table 10.** Land use pattern and carbon sink in Jilin Province in 2030.

|  | Cultivated Land | Forest Land | Grass Land | Water Body | Waste Land | Artificial Ground |
|---|---|---|---|---|---|---|
| Area (km$^2$) | 90,035.68 | 81,550.07 | 4535.75 | 3193.28 | 1411.198 | 10,245.79 |
| Carbon sink (tCO$_2$) | 165,100 | 17,492,500 | 35,000 | 86,600 | 2600 | 18,700 |

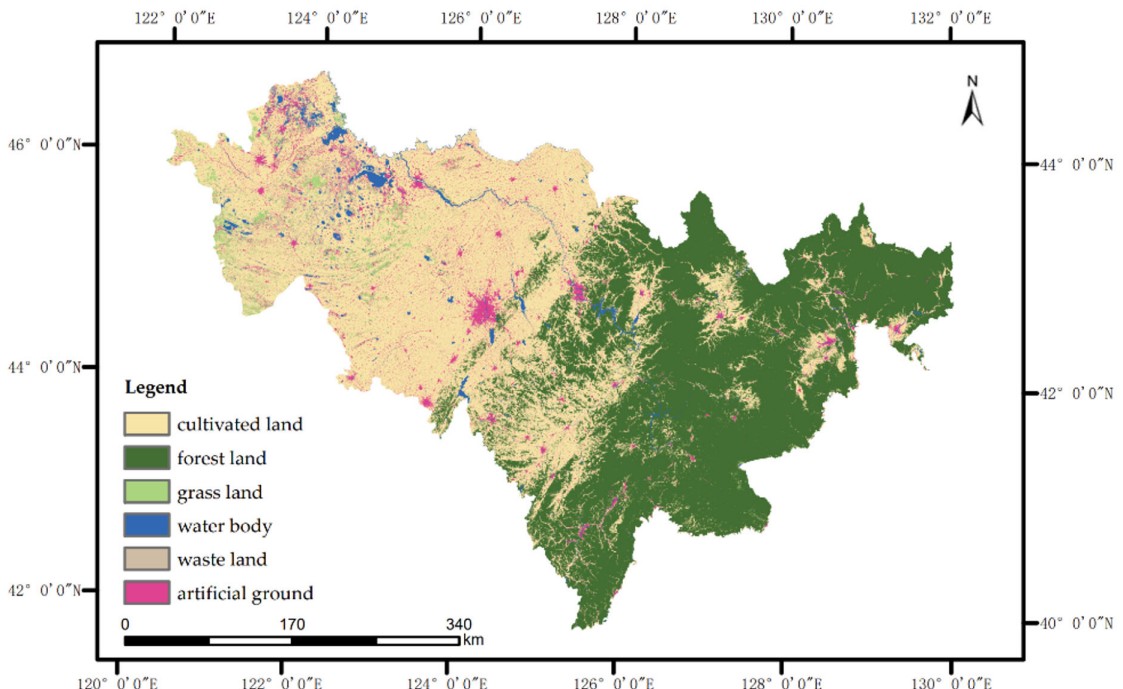

**Figure 4.** Projected land use structure of Jilin Province in 2030.

## 4. Discussion

The combination of natural factors and various socio-economic factors has contributed to significant changes in land use types and areas during the urbanization process. Natural factors have a more stable impact and do not change much over a long period, while cultivated land and artificial ground are more affected by the slope factor. Socio-economic factors are also the reasons that can lead to changes in land use types, including transportation roads, changes in urban and rural centers [35,36], the implementation of policies [37], and the adaptation atlas following this pattern. These factors also provide for an adaptive atlas. The influence of natural and human conditions on land use type changes has been proposed in areas such as Zhengzhou, China [38] and Algiers, Algeria [39].

### 4.1. Topographical Factors

Slope dominates the influence of topographic factors, and both the stability of the slope and the characteristics that limit the degree of land use development have the most direct impact on land use change. In this study, the slope is divided into six categories, ≤5°, 5–10°, 10–15°, 15–20°, 20–25°, and >25°, which are the six levels. Combined with Jilin Province, the land use type maps for 2011, 2014, 2017, and 2020 were superimposed on the slope map (Figure 5) to obtain the proportion of each land use type within different slopes in relation to its total area in the study area (Tables 11–14).

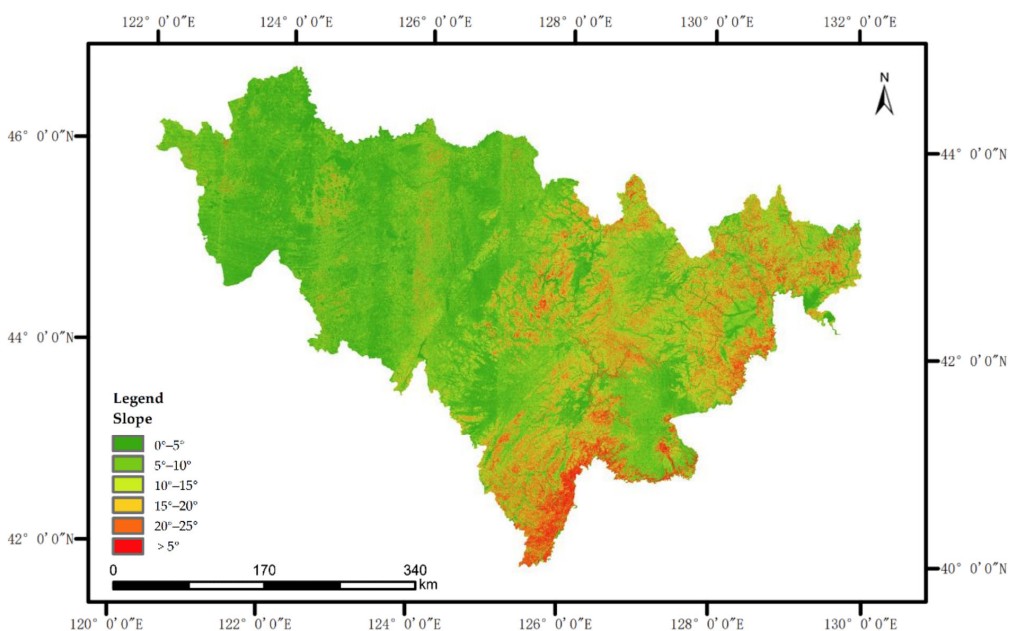

**Figure 5.** Topographic slope grade map of Jilin Province.

**Table 11.** Percentage of slope results for land use types in Jilin Province, 2011.

| Typology / Slope | Cultivated Land | Forest Land | Grass Land | Water Body | Waste Land | Artificial Ground |
|---|---|---|---|---|---|---|
| ≤5° | 50.00% | 13.74% | 59.12% | 89.33% | 72.93% | 63.25% |
| 5–10° | 34.00% | 24.15% | 31.56% | 7.77% | 23.75% | 29.19% |
| 10–15° | 11.03% | 23.41% | 6.85% | 1.96% | 2.88% | 5.80% |
| 15–20° | 3.35% | 18.16% | 1.61% | 0.59% | 0.36% | 1.24% |
| 20–25° | 1.06% | 11.07% | 0.47% | 0.21% | 0.05% | 0.32% |
| >25° | 0.56% | 9.49% | 0.38% | 0.14% | 0.02% | 0.18% |

**Table 12.** Percentage of slope results for land use types in Jilin Province, 2014.

| Typology / Slope | Cultivated Land | Forest Land | Grass Land | Water Body | Waste Land | Artificial Ground |
|---|---|---|---|---|---|---|
| ≤5° | 49.89% | 13.60% | 59.10% | 88.07% | 72.42% | 62.99% |
| 5–10° | 34.03% | 24.04% | 31.55% | 8.96% | 24.15% | 29.26% |
| 10–15° | 11.09% | 23.42% | 6.86% | 2.08% | 2.93% | 5.91% |
| 15–20° | 3.38% | 18.24% | 1.61% | 0.57% | 0.37% | 1.30% |
| 20–25° | 1.07% | 11.14% | 0.48% | 0.19% | 0.06% | 0.35% |
| >25° | 0.55% | 9.57% | 0.39% | 0.12% | 0.06% | 0.20% |

**Table 13.** Percentage of slope results for land use types in Jilin Province, 2017.

| Typology / Slope | Cultivated Land | Forest Land | Grass Land | Water Body | Waste Land | Artificial Ground |
|---|---|---|---|---|---|---|
| ≤5° | 50.00% | 13.63% | 58.65% | 88.57% | 72.63% | 62.83% |
| 5–10° | 33.97% | 24.07% | 31.67% | 8.51% | 23.99% | 29.28% |
| 10–15° | 11.05% | 23.40% | 7.02% | 2.03% | 2.89% | 5.98% |
| 15–20° | 3.37% | 18.22% | 1.71% | 0.58% | 0.37% | 1.33% |
| 20–25° | 1.06% | 11.13% | 0.52% | 0.19% | 0.06% | 0.36% |
| >25° | 0.55% | 9.56% | 0.43% | 0.12% | 0.06% | 0.21% |

**Table 14.** Percentage of slope results for land use types in Jilin Province, 2020.

| Typology <br> Slope | Cultivated Land | Forest Land | Grass Land | Water Body | Waste Land | Artificial Ground |
|---|---|---|---|---|---|---|
| ≤5° | 50.27% | 13.66% | 56.31% | 88.52% | 72.44% | 62.73% |
| 5–10° | 33.89% | 24.12% | 32.76% | 8.54% | 24.14% | 29.30% |
| 10–15° | 10.94% | 23.40% | 7.85% | 2.04% | 2.92% | 6.02% |
| 15–20° | 3.31% | 18.19% | 2.00% | 0.58% | 0.37% | 1.36% |
| 20–25° | 1.04% | 11.10% | 0.61% | 0.19% | 0.06% | 0.37% |
| >25° | 0.54% | 9.53% | 0.47% | 0.12% | 0.07% | 0.22% |

As can be seen from Tables 11–14, about 80 percent of the arable land is mainly distributed between 0 and 15°, in which the smaller the slope the larger the proportion of arable land; when the slope of the ground exceeds 15°, the proportion of the arable land area decreases sharply. Therefore, most of the arable land is distributed on a slope below 15°, and, according to the size of the slope, above 15° is a steep slope, which is not suitable for crop cultivation. Therefore, the conversion rule for arable land is that slopes below 15° are easily converted to arable land. Forest land is more evenly distributed on all slopes. To strictly protect the arable land and ensure reasonable land demand for social and economic development, the area with a slope greater than 15° is regarded as a suitable area for forest land conversion. Grassland, water bodies, wasteland, and artificial ground are mainly distributed between 0 and 10°, with a small amount between 10 and 15°. As the distribution of building land depends on human choice, areas with flat terrain are generally chosen for development and construction to save costs, and the higher the slope, the less the increase in the area of building land; building land increases between 0 and 15°. Grasslands, water bodies, and wastelands can be converted to any land use type, because they have fewer constraints.

*4.2. Road Factors*

The major transport road network has an impact on the change of land use structure in the region. Roads are the links connecting the neighborhood and various places and are the foundation and driving force of urban development. The planning and improvement of transport roads not only directly affects people's lives, but also has a significant impact on the change in land use structure in the process of urban development. Through the buffer analysis of the basic road network, four buffer zones (Figure 6) are established in units of 400 m for analysis (Tables 15–18).

**Table 15.** Area share of land use types in road network buffer zones in Jilin Province, 2011.

| Typology <br> Buffer | Cultivated Land | Forest Land | Grass Land | Water Body | Waste Land | Artificial Ground |
|---|---|---|---|---|---|---|
| <400 m | 18.61% | 5.51% | 0.53% | 0.23% | 0.09% | 5.28% |
| 400–800 m | 15.30% | 6.66% | 0.51% | 0.20% | 0.14% | 2.63% |
| 800–1200 m | 13.59% | 7.28% | 0.51% | 0.16% | 0.16% | 1.89% |
| 1200–1600 m | 11.76% | 7.01% | 0.53% | 0.23% | 0.11% | 1.07% |
| <1600 m | 59.25% | 26.47% | 2.08% | 0.82% | 0.50% | 10.88% |

**Table 16.** Area share of land use types in road network buffer zones in Jilin Province, 2014.

| Typology <br> Buffer | Cultivated Land | Forest Land | Grass Land | Water Body | Waste Land | Artificial Ground |
|---|---|---|---|---|---|---|
| <400 m | 16.62% | 6.84% | 0.56% | 0.27% | 0.13% | 5.59% |
| 400–800 m | 14.62% | 7.06% | 0.54% | 0.13% | 0.19% | 2.69% |
| 800–1200 m | 14.19% | 7.21% | 0.43% | 0.24% | 0.20% | 1.94% |
| 1200–1600 m | 11.93% | 6.58% | 0.44% | 0.21% | 0.08% | 1.29% |
| <1600 m | 57.36% | 27.69% | 1.97% | 0.86% | 0.60% | 11.51% |

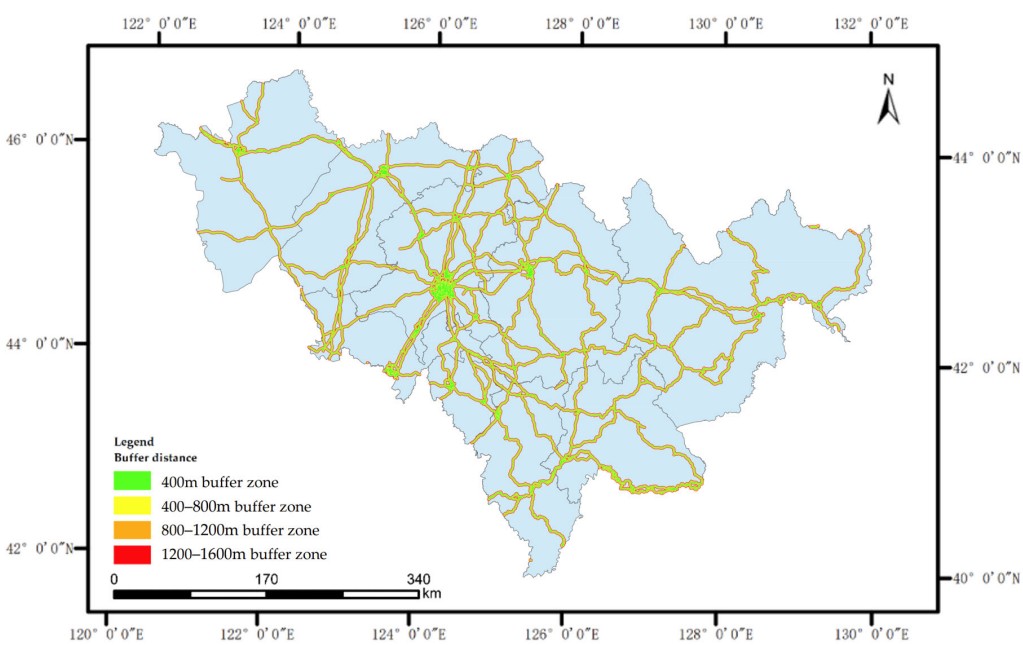

**Figure 6.** Diagram of the road buffer zone in Jilin Province.

**Table 17.** Area share of land use types in road network buffer zones in Jilin Province, 2017.

| Typology<br>Buffer | Cultivated<br>Land | Forest<br>Land | Grass<br>Land | Water<br>Body | Waste<br>Land | Artificial<br>Ground |
|---|---|---|---|---|---|---|
| <400 m | 16.32% | 6.84% | 0.56% | 0.28% | 0.11% | 5.90% |
| 400–800 m | 14.45% | 7.05% | 0.50% | 0.13% | 0.19% | 2.92% |
| 800–1200 m | 14.11% | 7.21% | 0.44% | 0.23% | 0.20% | 2.02% |
| 1200–1600 m | 11.86% | 6.62% | 0.39% | 0.23% | 0.09% | 1.34% |
| <1600 m | 56.75% | 27.72% | 1.89% | 0.87% | 0.59% | 12.18% |

**Table 18.** Area share of land use types in road network buffer zones in Jilin Province, 2020.

| Typology<br>Buffer | Cultivated<br>Land | Forest<br>Land | Grass<br>Land | Water<br>Body | Waste<br>Land | Artificial<br>Ground |
|---|---|---|---|---|---|---|
| <400 m | 15.87% | 6.92% | 0.56% | 0.29% | 0.12% | 6.25% |
| 400–800 m | 14.27% | 7.12% | 0.44% | 0.13% | 0.19% | 3.08% |
| 800–1200 m | 14.11% | 7.22% | 0.40% | 0.21% | 0.13% | 2.13% |
| 1200–1600 m | 11.73% | 6.63% | 0.43% | 0.21% | 0.12% | 1.41% |
| <1600 m | 55.98% | 27.89% | 1.84% | 0.86% | 0.56% | 12.87% |

Between 2011 and 2020, the area of cultivated land within the buffer zone of major traffic roads of 1600m shows a decreasing trend; the proportion of cultivated land decreases with the increase in the distance from the buffer zone, but the area of cultivated land within the buffer zone is dominant, accounting for more than 50%, and the proximity to the road determines the trend in the change in cultivated land. The main traffic routes have a strong attraction effect on the construction land and the roads have a positive influence on the increase in construction land area; the closer the roads are, the larger the proportion of construction land development. The 400m buffer zone has the largest proportion of artificial ground occupation, and the proportion of land occupation decreases with the increase in distance. Along with the promotion of the process of urbanization and the steady development of the economy, the influence of the traffic roads on land use development is becoming more and more significant. The proportion of forest land, grassland, water body, and wasteland within the buffer zone is very little affected by changes in distance.

### 4.3. Township Centers

The township center also plays a crucial role in the development, creating a buffer zone with a radius of 1000 m (Figure 7), establishing five buffer zones, and overlaying the buffer zones with the current land use status maps of 2011, 2014, 2017 and 2020, respectively. The percentage of the area of each land use type in the overlapping part is obtained (Tables 19–22).

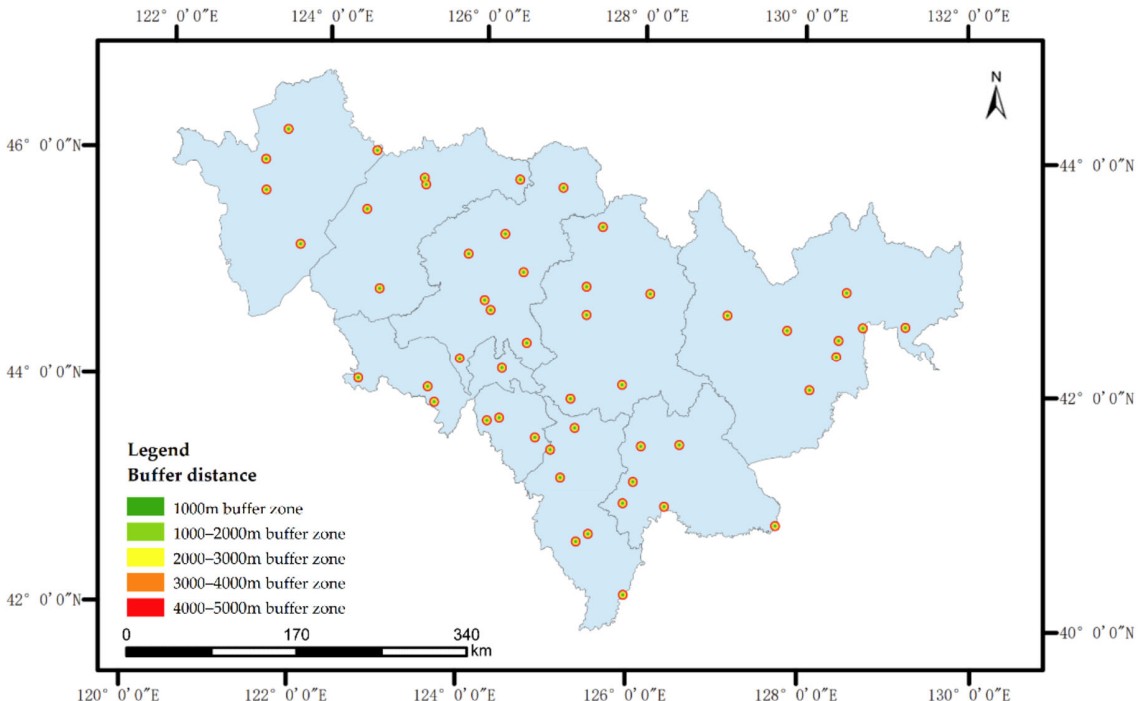

**Figure 7.** Diagram of the buffer zone of the township in Jilin Province.

**Table 19.** Area share of land use types in the buffer zone of townships in Jilin Province, 2011.

| Typology Buffer | Cultivated Land | Forest Land | Grass Land | Water Body | Waste Land | Artificial Ground |
|---|---|---|---|---|---|---|
| <1 km | 0.55% | 0.24% | 0.00% | 0.03% | 0.00% | 3.29% |
| 1–2 km | 3.91% | 1.14% | 0.02% | 0.24% | 0.02% | 6.77% |
| 2–3 km | 9.72% | 3.20% | 0.10% | 0.38% | 0.09% | 6.42% |
| 3–4 km | 15.92% | 5.73% | 0.25% | 0.32% | 0.18% | 5.47% |
| 4–5 km | 21.59% | 8.27% | 0.43% | 0.40% | 0.28% | 5.04% |
| <5 km | 51.68% | 18.58% | 0.79% | 1.37% | 0.57% | 27.00% |

**Table 20.** Area share of land use types in the buffer zone of townships in Jilin Province, 2014.

| Typology Buffer | Cultivated Land | Forest Land | Grass Land | Water Body | Waste Land | Artificial Ground |
|---|---|---|---|---|---|---|
| <1 km | 0.48% | 0.23% | 0.00% | 0.02% | 0.00% | 3.37% |
| 1–2 km | 3.52% | 1.13% | 0.02% | 0.24% | 0.01% | 7.18% |
| 2–3 km | 9.19% | 3.19% | 0.09% | 0.37% | 0.08% | 6.98% |
| 3–4 km | 15.38% | 5.71% | 0.23% | 0.31% | 0.16% | 6.07% |
| 4–5 km | 21.07% | 8.21% | 0.36% | 0.42% | 0.27% | 5.66% |
| <5 km | 49.65% | 18.47% | 0.71% | 1.37% | 0.53% | 29.27% |

**Table 21.** Area share of land use types in the buffer zone of townships in Jilin Province, 2017.

| Typology<br>Buffer | Cultivated<br>Land | Forest<br>Land | Grass<br>Land | Water<br>Body | Waste<br>Land | Artificial<br>Ground |
|---|---|---|---|---|---|---|
| <1 km | 0.45% | 0.24% | 0.00% | 0.03% | 0.00% | 3.39% |
| 1–2 km | 3.34% | 1.14% | 0.02% | 0.23% | 0.01% | 7.37% |
| 2–3 km | 8.86% | 3.22% | 0.09% | 0.38% | 0.06% | 7.31% |
| 3–4 km | 15.02% | 5.75% | 0.21% | 0.30% | 0.13% | 6.44% |
| 4–5 km | 20.73% | 8.25% | 0.35% | 0.41% | 0.23% | 6.03% |
| <5 km | 48.41% | 18.60% | 0.67% | 1.35% | 0.43% | 30.55% |

**Table 22.** Area share of land use types in the buffer zone of townships in Jilin Province, 2020.

| Typology<br>Buffer | Cultivated<br>Land | Forest<br>Land | Grass<br>Land | Water<br>Body | Waste<br>Land | Artificial<br>Ground |
|---|---|---|---|---|---|---|
| <1 km | 0.43% | 0.24% | 0.00% | 0.03% | 0.00% | 3.41% |
| 1–2 km | 3.19% | 1.15% | 0.02% | 0.24% | 0.01% | 7.49% |
| 2–3 km | 8.54% | 3.26% | 0.09% | 0.39% | 0.05% | 7.59% |
| 3–4 km | 14.62% | 5.82% | 0.20% | 0.31% | 0.11% | 6.80% |
| 4–5 km | 20.43% | 8.35% | 0.28% | 0.42% | 0.17% | 6.35% |
| <5 km | 47.22% | 18.82% | 0.60% | 1.39% | 0.34% | 31.64% |

The share of arable land area is decreasing in the period 2011–2020, while the share of woodland and man-made surface area is significantly increasing. The proportion of the artificial ground surface area is expanding. It can be seen that the increase in the area of the artificial ground surface in the buffer zone is related to the decrease in the area of arable land and the conversion of land use types. The share of wasteland and grassland areas has slightly decreased. As the distance from the township center increases, the proportion of cultivated land and forested land increases significantly, while the proportion of artificial ground surface area gradually becomes smaller.

*4.4. Socio-Economic Drivers*

Socio-conomic development cannot be separated from the positive effects of population growth, and rapid economic development and population growth will lead to changes in land use patterns in socio-economic activities. Land for housing, transportation, and services will expand as the population grows, and the supply of food will require an increase in the amount of cultivated land. The increase in the area of land for construction and cultivated land will lead to changes in other land use types, resulting in changes in the structure of land use types.

*4.5. Precision Comparison*

After IDRISI's Crosstab module, the 2020 land simulation prediction maps with the adaptive atlas as a condition were spatially overlaid with the 2020 status quo maps (Table 23), and Kappa coefficients were obtained to achieve the accuracy validation result, which was 0.963.

**Table 23.** Error analysis of land use type simulation and forecast of the study area.

| Land Use Type | Status Data<br>for 2020 (km$^2$) | Modelled Area<br>in 2020 (km$^2$) | Relative Error |
|---|---|---|---|
| cultivated land | 89,555.7 | 90,538.2 | −0.01 |
| forest land | 82,320.8 | 81,004.8 | 0.02 |
| grass land | 5136.9 | 5065.3 | 0.02 |
| water body | 2782.9 | 2886.1 | −0.04 |
| waste land | 1917.8 | 1862.2 | 0.29 |
| artificial ground | 9071.1 | 9524.5 | −0.05 |

The accuracy of land use types in Dohuk Governorate, Iraq [40] predicted by CA–Markov model in 2017 (Table 24); the Kappa coefficient is 0.918.

**Table 24.** Accuracy assessment (error matrix) for 2017.

| Class | Dense Forest | Sparse Forest | Agricultural Land | Urban Area | Barren Area | Water Body | Total | User Accuracy |
|---|---|---|---|---|---|---|---|---|
| Dense forest | 36 | 3 | 0 | 0 | 0 | 0 | 39 | 0.923 |
| Sparse forest | 2 | 37 | 0 | 0 | 0 | 0 | 39 | 0.949 |
| Agricultural land | 0 | 0 | 37 | 0 | 2 | 0 | 39 | 0.949 |
| Urban area | 0 | 0 | 0 | 35 | 4 | 0 | 39 | 0.897 |
| Barren area | 0 | 0 | 4 | 0 | 35 | 0 | 39 | 0.897 |
| Water body | 0 | 0 | 1 | 0 | 0 | 38 | 39 | 0.974 |
| Total | 38 | 40 | 42 | 35 | 41 | 38 | 234 | 0 |
| Producer accuracy | 0.947 | 0.925 | 0.881 | 1 | 0.854 | 1 | 0 | 0.932 |
| Kappa | 0.918 | | | | | | | |

After comparison, the predicted values will be more accurate and have higher Kappa coefficients within the limitations of the influence of natural and social factors.

*4.6. Relationship between Land Use and Sustainable Development*

As the main carrier of human–nature interaction, the land use system plays an important role in sustainable development in the region. The use of land as a resource is the foundation of sustainable development. Simultaneously, land use can directly influence the transformation of carbon sinks and, by extension, the transformation of the ecological environment. (1) Effects on the regional forest. The ecological environment is in jeopardy, due to the disappearance of forestland. Influenced by economic growth, population increase, and urbanization, the forest land area in Jilin Province has been significantly reduced and the urban area has been expanding. The reduction in forest land will break the foundation of regional development. (2) Regional environmental consequences. Various terrestrial ecosystems are dispersed throughout the territory. Different ecosystem types, locations, and spatial distribution patterns will alter as a result of changes in land use. Massive changes in land use have a direct effect on the diversity and magnitude of ecosystem services [41]. In 2011–2020, forest and grassland areas in Jilin province have been decreasing, reducing the natural and ecological purification functions. The felling of trees significantly diminishes the function of water resources in conserving water. Therefore, the change in land use will harm the regional ecological environment and threaten the region's sustainable growth. (3) Influence on regional socio-economic development. The decline in the quality of the regional ecological environment will limit socio-economic growth, resulting in a situation where both the socio-economic and ecological environment lag [42]. To set a solid foundation for regional sustainable development, it is necessary to establish a sustainable land use model.

This study did not consider the changing pattern of woodland and grassland from the point of view of climatic factors; for example, drought leads to a decrease in the growth rate of plants and trees, an increase in wildfires, or an increase in growth due to an increase in rainfall and temperature. The effect of climatic factors should be added to the next study to make the data more accurate and reliable.

**5. Conclusions**

Based on the land use data of Jilin Province in 2011, 2014, 2017, and 2020, land use changes as well as carbon sink changes in this range were analyzed and predicted, and the following conclusions were drawn:

(1)    Cultivated land and forest land are the two most prevalent land uses in Jilin Province, and their areas are close to 90% of the total area of the province. From 2011 to 2020, the cultivated agricultural area, water body, and artificial ground in the province

increased. The region of artificial ground continued to increase, with an increase of 1406.85 km$^2$, while the area of grassland decreased most significantly, with a decrease of 1984 km$^2$.

(2)  The increased area of artificial ground is mainly converted from cropland, accounting for 70.34% of the total converted area. The area of forest land is mainly converted with the area of cropland, and grassland is mainly converted to cropland area, accounting for 84.96% of the total converted area. The water body and wasteland are mainly converted to cultivated land and artificial ground, and the area of artificial ground undergoing conversion is smaller.

(3)  During 2011–2020, the change in carbon sink mainly comes from forest land carbon sink and grassland carbon sink, decreasing by 59,200 tCO$_2$ and 15,200 tCO$_2$, respectively, while the carbon sink of other land use types does not fluctuate much. The total carbon sink decreases by 66,000 tCO$_2$.

(4)  In 2030, compared with 2020, the area of woodland, grassland, and wasteland and their corresponding carbon sinks are peredicted to decrease, with the area and carbon sinks of woodland decreasing the most, by 770.74 km$^2$ and 165,300 tCO$_2$, respectively. The carbon sinks of cropland, water body, and artificial ground increase slightly. The overall carbon sink was 17,805,000 tCO$_2$, a decrease of 156,700 tCO$_2$ from 2020.

**Author Contributions:** Conceptualization, M.W. and C.D.; methodology, M.W.; software, M.W.; validation, M.W.; resources, C.D.; supervision, X.W.; funding acquisition, X.W. All authors have read and agreed to the published version of the manuscript.

**Funding:** This study was funded by [Jilin Provincial Department of Education] under grant number JJKH20230724KJ.

**Institutional Review Board Statement:** Not applicable.

**Informed Consent Statement:** Not applicable.

**Data Availability Statement:** The data that support the findings of this study are available from the corresponding author on reasonable request.

**Conflicts of Interest:** The authors declare no conflict of interest.

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
