# Peer review of "Analysis and Forecast of Land Use and Carbon Sink Changes in Jilin Province, China"

_sustainability, doi:10.3390/su151914040_

Round 1

Reviewer 1 Report (Previous Reviewer 1)

As for the revised version, most suggestions have been revised. In the Line 199, (3) Combination of CA and Markov, I suggest put forward some equations for the CA and Markov, which is needed.

Quality of English can be improved.

Author Response

Some equations are provided for CA and Markov models.

Reviewer 2 Report (Previous Reviewer 2)

The authors have added a lot of tables of data, but what they have not added and which I still think is necessary is any discussion on error and uncertainty. I suspect, that some of the apparent changes are just due to how the data was collected and how land use categories are defined, but neither of these is discussed; for example, how certain is it that a parcel of land is grassland or waste land? I think the authors need to discuss data quality.

Information may not exist, but, if it does, I would be interested in a line or two comparing changes in this Province to changes in other Provinces.

Need to spell out CA as cellular automata earlier, it is finally spelled out on about line 150 which is too far into the paper for a first definition)

Addition on lines 81-83 can be deleted the earlier insertion is enough.

No range of values in table 1.

Table 9 - are these really transition probabilities, is there really a 13% probability that cultivated land will become forest?

line 326 - error in referencing.

tables 15, 16, 17 would I think be better if the last line of data was >1600 not <1600 similarly for tables 19 to 22 >5km not <5km

Author Response

2.81-83 rows were deleted, Table 1 was modified, 326 rows were deleted, CA with Markov model was defined earlier, Table 15-22 shows the effect of roads and urban-rural centers, at >1600m vs >5km, the effect becomes small, so < was used, and some of the cultivated land will be turned into forests due to the environmental policy and the migration of rural population. The article used ARCGIS with ENVI for classification and identification.

Reviewer 3 Report (Previous Reviewer 3)

Dear Authors 

Thanks for the replay to the queries.

Page no 2 lines 80-82: before and after the release of the plan, that could be minimized.

-

Author Response

Lines 80-82 on the second page were deleted.

This manuscript is a resubmission of an earlier submission. The following is a list of the peer review reports and author responses from that submission.

Round 1

Reviewer 1 Report

 This manuscript focuses on land use changes and carbon sink changes and selected the Jilin Province of China as the study area. Based on the land use data changes in Jilin Province in 2011, 2014, 2017 and 2020, the manuscript analyzed the land use changes from 2011 to 2020 using land transition matrix, calculates the carbon sink changes in recent years, predicted the land use types and carbon sink changes in Jilin Province in 2030 by applying the CA-Markov model, and explores the driving factors of land use changes. About this topic is currently a hot topic in this field. However, the manuscript needs majo revision and check before publication.

 Here are some specific suggestions for the authors.

 1. The language issues in this manuscript should be further polished by a native speaker or author’s colleague.

 2. In Introduction Section, the topic of China's carbon peaking and carbon neutrality targets should be involved. This manuscript talks about what it means for them. What is more, it seem that there is no a scientific question in the Introduction Section, which is very important.

 3. There are errors in Figure 1. First, there are no national boundaries; second, there is no South China Sea; third, the map should not be in too light a color, resulting in missing white parts. The letters symbol the longitude and latitude is too small to read.

 4. In Section 2.2, why was land use data from four points in time 2011, 2014, 2017, and 2020 used for the analysis. In addition, what products or what kind of method for the data and what is the accuracy from this data center? It is recommended that literatures be added for this point.

 5. In 2.3 Section “Land use transfer matrix”, the net rate of change was introduced. In essence, it is the prevalent single land use dynamic degree. What are the reasons and why Equation 3 is used here? Why do authors do not use the dynamic degree model? Readers might know this reason. On the other hand, some researchers have questioned about the popular dynamic degree model and land transition matrix. Therefore, it is suggested that authors should adopt any level of intensity analysis methodology for this research. Specifically, author can read as follows:

l  Huang, B., Huang, J., Pontius Jr, R. G., Tu, Z., (2018). Comparison of Intensity Analysis and the land use dynamic degrees to measure land changes outside versus inside the coastal zone of Longhai, China. Ecological Indicators, 89, 336-347. doi: 10.1016/j.ecolind.2017.12.057

l  Deng, Z., Quan, B., (2023). Intensity Analysis to Communicate Detailed Detection of Land Use and Land Cover Change in Chang-Zhu-Tan Metropolitan Region, China. Forests, 14 (5), 939-957. doi: 10.3390/f14050939

l  Aldwaik, S. Z., Pontius Jr, R. G., (2012). Intensity analysis to unify measurements of size and stationarity of land changes by interval, category, and transition. Landscape and Urban Planning, 106 (1), 103-114. doi: 10.1016/j.landurbplan.2012.02.010

l  Quan, B., Pontius Jr, R. G., Song, H., (2020). Intensity Analysis to communicate land change during three time intervals in two regions of Quanzhou City, China. GIScience & Remote Sensing, 57 (1), 21-36. doi: 10.1080/15481603.2019.1658420

l  Akodéwou, A., Oszwald, J., Saïdi, S., Gazull, L., Akpavi, S., Akpagana, K., Gond, V., (2020). Land Use and Land Cover Dynamics Analysis of the Togodo Protected Area and Its Surroundings in Southeastern Togo, West Africa. Sustainability, 12 (13). doi: 10.3390/su12135439

 6.  In Section 2.6, the introduction to the CA-Markov model and the associated parameter determination is too limited. This is not suitable for readers who want to understand the research methodology.

 7. The font size of the latitude, longitude and legend in Figure 2 is too small to be clear for the readers. It is recommended that Figure 2 be made clearer.

8. The title of the vertical axis in Figure 3 has been overlapped with other figures, please revise this. Also the second is aero, so it can be cancel in the Figure 3.

 9. In the Table 3, we find that 1238.86, -275.92, -1984.00,171.47, -554.89, 1406.85 is repeated so we doubt that whether the "variation" used in Table 3? Please check.

10. Please check the units of carbon sinks in Table 10.

11. In the Discussion Section, the drivers of land use change in Jilin Province were analyzed in terms of topography, roads, location of government, and socio-economic perspectives. However, there is no substance data and figure to support the driver analysis. This is not appropriate. A major revision is recommended here. For example, for the analysis of the effect of altitude on land use change, the authors could analyze what land types dominate or what kind of land conversion processes occur at different elevations.

12. Please pay attention to the citation format of reference.

 This manuscript focuses on land use changes and carbon sink changes and selected the Jilin Province of China as the study area. Based on the land use data changes in Jilin Province in 2011, 2014, 2017 and 2020, the manuscript analyzed the land use changes from 2011 to 2020 using land transition matrix, calculates the carbon sink changes in recent years, predicted the land use types and carbon sink changes in Jilin Province in 2030 by applying the CA-Markov model, and explores the driving factors of land use changes. About this topic is currently a hot topic in this field. However, the manuscript needs majo revision and check before publication.

 Here are some specific suggestions for the authors.

 1. The language issues in this manuscript should be further polished by a native speaker or author’s colleague.

 2. In Introduction Section, the topic of China's carbon peaking and carbon neutrality targets should be involved. This manuscript talks about what it means for them. What is more, it seem that there is no a scientific question in the Introduction Section, which is very important.

 3. There are errors in Figure 1. First, there are no national boundaries; second, there is no South China Sea; third, the map should not be in too light a color, resulting in missing white parts. The letters symbol the longitude and latitude is too small to read.

 4. In Section 2.2, why was land use data from four points in time 2011, 2014, 2017, and 2020 used for the analysis. In addition, what products or what kind of method for the data and what is the accuracy from this data center? It is recommended that literatures be added for this point.

 5. In 2.3 Section “Land use transfer matrix”, the net rate of change was introduced. In essence, it is the prevalent single land use dynamic degree. What are the reasons and why Equation 3 is used here? Why do authors do not use the dynamic degree model? Readers might know this reason. On the other hand, some researchers have questioned about the popular dynamic degree model and land transition matrix. Therefore, it is suggested that authors should adopt any level of intensity analysis methodology for this research. Specifically, author can read as follows:

l  Huang, B., Huang, J., Pontius Jr, R. G., Tu, Z., (2018). Comparison of Intensity Analysis and the land use dynamic degrees to measure land changes outside versus inside the coastal zone of Longhai, China. Ecological Indicators, 89, 336-347. doi: 10.1016/j.ecolind.2017.12.057

l  Deng, Z., Quan, B., (2023). Intensity Analysis to Communicate Detailed Detection of Land Use and Land Cover Change in Chang-Zhu-Tan Metropolitan Region, China. Forests, 14 (5), 939-957. doi: 10.3390/f14050939

l  Aldwaik, S. Z., Pontius Jr, R. G., (2012). Intensity analysis to unify measurements of size and stationarity of land changes by interval, category, and transition. Landscape and Urban Planning, 106 (1), 103-114. doi: 10.1016/j.landurbplan.2012.02.010

l  Quan, B., Pontius Jr, R. G., Song, H., (2020). Intensity Analysis to communicate land change during three time intervals in two regions of Quanzhou City, China. GIScience & Remote Sensing, 57 (1), 21-36. doi: 10.1080/15481603.2019.1658420

l  Akodéwou, A., Oszwald, J., Saïdi, S., Gazull, L., Akpavi, S., Akpagana, K., Gond, V., (2020). Land Use and Land Cover Dynamics Analysis of the Togodo Protected Area and Its Surroundings in Southeastern Togo, West Africa. Sustainability, 12 (13). doi: 10.3390/su12135439

 6.  In Section 2.6, the introduction to the CA-Markov model and the associated parameter determination is too limited. This is not suitable for readers who want to understand the research methodology.

 7. The font size of the latitude, longitude and legend in Figure 2 is too small to be clear for the readers. It is recommended that Figure 2 be made clearer.

8. The title of the vertical axis in Figure 3 has been overlapped with other figures, please revise this. Also the second is aero, so it can be cancel in the Figure 3.

 9. In the Table 3, we find that 1238.86, -275.92, -1984.00,171.47, -554.89, 1406.85 is repeated so we doubt that whether the "variation" used in Table 3? Please check.

10. Please check the units of carbon sinks in Table 10.

11. In the Discussion Section, the drivers of land use change in Jilin Province were analyzed in terms of topography, roads, location of government, and socio-economic perspectives. However, there is no substance data and figure to support the driver analysis. This is not appropriate. A major revision is recommended here. For example, for the analysis of the effect of altitude on land use change, the authors could analyze what land types dominate or what kind of land conversion processes occur at different elevations.

12. Please pay attention to the citation format of reference.

Reviewer 2 Report

There are numerous problems with missing references in the text, eg lines, 36, 47, 48, 56, 66, 67 & 68 These are problems that also exist in the "non-published material version" which was, presumably, an earlier draft sent out for review and not as the title suggests additional of more detailed data.

It is not clear to me what is meant by "waste" in the context of land use in this region of China, a more descriptive name should be used with a brief explanation of what is meant.

The time period covered and the prediction period is quite short but I was surprised there was no mention of climate change. Even small changes in climate might increase or decrease carbon sequestration by significant amounts; e.g. drought leading to lower growth rate of plants or increased wild fire; or increased rainfall and temperature leading to increased growth (or increased decay).

The primary problem I have with this paper is that there is no discussion of error. Because there is no discussion of error I have no idea whether the "changes" are real or just artifacts of the way the data was produced. The fact that the land cover is in 30 meter pixels leads me to suspect that the land cover is derived from Landsat imagery, which further leads me to suspect that the miss-classification error could exceed the rate of change. Therefore more detail is needed on what the base data is.

It has been some years since I've used Idrisi but based on my experience I have some confidence the model will be reasonably robust, but, the MC part must be run to include error and uncertainty.

Reviewer 3 Report

Dear Authors

I appreciate your interest in the assessment of land use and carbon sink change in Jilin Province, China. Unfortunately, it is a very basic assessment and needs a detailed analysis along with field investigation.

Lacking Corban sink-related literature in the introduction. This section fully discussed only lulc.

Need to frame a clear objective of the research work.

Why, the authors are selected these years (2011,2014,2017,2020) for the analysis?

The authors could provide a detailed explanation and procedure for CA-Markov in the methodology section.

Very weak discussion section. Hope those supporting suitability factors and their maps could be shifted to the methodology/result section. The discussion should be logical and a comparative assessment.

There are several spell errors and typographic errors.

Need a major language correction
